# Influence of Inertia on the Dynamic Compressive Strength of Concrete

**DOI:** 10.3390/ma15207278

**Published:** 2022-10-18

**Authors:** Zhangchen Qin, Dan Zheng, Xinxin Li, Haicui Wang

**Affiliations:** School of River and Ocean Engineering, Chongqing Jiaotong University, Chongqing 400074, China

**Keywords:** inertia effect, strain rate, strength, dynamic increase factor, concrete

## Abstract

The rate sensitivity of concrete material is closely related to the inertia and viscous effects. However, the effect of inertia on the dynamic strength of concrete remains unclear. In this paper, digital image correlation technology was applied to study the strain variation of dry and saturated concrete with different loading rates. The test results indicated that the strain gradually decreased with the distance from the load end, and the strain gradient around the load region increased with the strain rate, especially for saturated concrete. Then, a single degree of freedom model was established to evaluate the dynamic compressive strength of elastic concrete. The calculated results indicated that the influence of inertia on the dynamic increase factor (DIF) was negligible for concrete within a low strain rate. When the strain rate is larger than 10^0^/s, the inertial effect on the strength of concrete should be considered. After that, a quasi-static concrete damaged plasticity (CDP) model was employed to simulate the influence of inertia on the stress distribution and axial reaction force at the loaded end of concrete under different rates of compressive loading and verified with experimental results. The results obtained in this study indicated that the dynamic nominal strength of concrete obtained from the tests could not be directly used for structural analysis which may overestimate the effect of inertia on the dynamic response of the structure.

## 1. Introduction

During the service life of concrete structures, the properties of concrete will be significantly influenced by seismic blasts or impact loading with high loading rates [1,2,3,4]. To accurately evaluate the dynamic response of concrete structures under these extreme dynamic loading conditions, Abrams [5] first performed a series of compression tests on concrete with a strain rate of 2 × 10^−4^ s^−1^ and found that the compressive strength of concrete is an increasing function of the strain rate. Later, various experimental studies were carried out to quantify the effect of loading rate on the dynamic performance of concrete. Owing to the mechanical properties, geometric dimensions, and curing conditions of the specimens and the test equipment being quite different, DIF (the ratio of measured dynamic concrete strength to static strength during the test) was adopted to identify the effect of strain rate on the strength of concrete. The test results showed that the strength of concrete increases with the strain rate, and the rate sensitivity of concrete with low strength is larger than that with high strength. As the strain rate is increased to a value larger than 10^0^ s^−1^~10^1^ s^−1^, a sudden increase in strength can be observed [1,2,3,4,5]. 

Many studies have been conducted to investigate the mechanism of the strain rate on the strength of concrete. For dry concrete, Rossi et al. [6,7,8] observed that, in typical concrete strength tests, the initial state of concrete is static. As the dynamic loading is applied to the concrete with a high loading rate, under the influence of inertia, the specimen cannot be changed from the static to the dynamic state immediately. Therefore, the inertial effect can be regarded as the main factor influencing the dynamic strength of concrete. Eibl et al. [9] and Bischoff et al. [1,10] further pointed out that a higher loading rate resulted in a larger rate sensitivity of concrete. Based on the visco-plastic model, Georgin and Reynouard [11] also found that the inertial force and dynamic structural effect of the specimen exerted an influence on the dynamic strength of concrete. Donze et al. [12] confirmed the contribution of the radial inertia confinement effect on the rate sensitivity of concrete compressive strength by discrete element numerical simulation. After that, many researchers [2,13,14,15,16,17,18,19,20,21,22] conducted numerical and experimental studies on the dynamic compressive strength test of concrete and found that the rate sensitivity of concrete was not only related to the properties of concrete, but the radial inertial confining pressure effect should also be considered. However, the radial inertial confining effect cannot explain the rate effect of concrete tensile strength [23,24,25,26]. Furthermore, a few scholars concluded that concrete is a rate-independent material. The rate sensitivity measured by the dynamic tests is caused by the inertial effect [27,28], and it is recommended that the static strength of concrete is used for dynamic structural analysis. For wet concrete, with the increasing moisture content, the strength improvement becomes more sensitive to the variation of the loading rate than dry concrete. Rossi et al. [8] and Zheng et al. [29] attributed the rate sensitivity of wet concrete to the viscosity of free and chemically combined water. The chemically combined water is an integral part of the hydrated cement paste and could not evaporate on drying. Therefore, the rate effect of concrete is related to the existence of inertial and viscous forces under different strain rates. Moreover, it is worth noting that the inertia effect has been considered in the structural dynamic equations, such that the strength value obtained in the tests cannot be directly used as the dynamic concrete strength, and the correct material properties must be given based on an accurate analysis of the inertia effect. However, the information on this topic is limited. 

Therefore, the objective of this paper was to investigate the mechanism of inertia on the dynamic strength of concrete. Based on the strain distribution in concrete during the compressive tests, the relationship between dynamic nominal strength and actual strength of concrete was analyzed. Then, the effect of inertia on the nominal strength and actual strength of concrete under different strain rates was evaluated by a single degree of freedom model. Finally, a finite element numerical model based on the static test parameters was established and verified with the test results.

## 2. Experimental Procedures

### 2.1. Specimen Preparation

According to the test code for hydraulic concrete (SL/T 352-2020), 8 specimens with dimensions of 150 mm × 150 mm × 150 mm were adopted in the tests to study the strain variation of dry and saturated concrete under different loading rates. Therefore, one specimen of each test condition was sprayed with irregular speckles before testing. The maximum aggregate size of concrete was 20 mm, and the mix of proportions by weight was 1: 0.50: 1.48: 2.75 (cement: water: sand: aggregate). The specimens were cured for 28 days after casting. The specimens adopted in this paper can be divided into dry and saturated specimens. For dry specimens, the cured specimens were placed in an oven at the temperature of 90 °C for around 15 days until the weight of the specimens was not changed with the drying time. However, for saturated specimens, the specimens were placed in water until the weight of the specimens was not changed with the immersion time. After the specimens were dry and saturated enough, irregular speckles were sprayed on the testing surface.

### 2.2. Test Produce

As shown in Figure 1, the test was conducted on a microcomputer-controlled automatic pressure testing machine with a maximum pressure of 3000 kN. In order to study the mechanical properties of dry and saturated specimens at different strain rates, four strain rates were adopted by displacement control in this paper, and the specimen numbers and loading rates are listed in Table 1. The pressure *P* and the loading time *t* of the tests were recorded by the machine system. A CCD camera was set up in front of the specimens to record the surface strain of the specimens under different pressures. The resolution of the camera was 2048 × 2048 and the frame rate was adjusted to fit different loading rates. The VIC-2D system was adopted in the test with the accuracy of 10 uε and developed by the Correlated Solutions INC, USA. Before testing, the accuracy of this VIC-2D system was verified by the strain gauge with an accuracy of 0.5 uε. Tests showed that the error between DIC technology and the strain gauge was within 3%. The specimens were preloaded to 2 kN before the test, and then monotonically loaded till failure.

## 3. Results and Discussion

Figure 2 shows the principal strain cloud map of D1 at different loading stages, when the compression load attained 20% to 60% *F_max_* (the max value of pressure during loading), the principal strain presented a decreasing trend from the loading end to the fixed end. When the load increased to 80% *F_max_*, strain concentration appeared on the right side of the surface, and a part of the cloud map is missing because of the concrete surface peeling off. When the stress attained the peak loading, a large area of stress concentration appeared on the surface of the specimen and some visible cracks could also be observed.

As shown in Figure 3, to further study the inertia and viscous effects on the loading process, line AB located in the middle of the specimen was selected to describe the variation of *ε_yy_* (vertical strain) under different loading stages. For the D1 specimen, the vertical strain gradually decreased from the loading end to the fixed end, and when the load was between 20% and 80% *F_max_*. The value of *ε_yy_* increased with the compressive loading, and the overall shape of the *ε_yy_* curves at different loading stages presented similarly. It is worth noting that when the load increased to 90% *F_max_*, the strain concentration appeared around 10~20 mm height of the D1 specimen. The vertical strain at the same point and the time of the strain concentration gradually decreased with the strain but the overall shape of the strain distribution was not significantly changed. Owing to the specimens D1 to D4 being dried before testing, the effect of viscosity on the compressive response of concrete can be neglected. Therefore, the test results indicated that the decreasing trend of the vertical strain even under different strain rates can be attributed to the effect of inertia. 

Figure 3c illustrates the variation of *ε_yy_* for saturated specimens under different strain rates. It can be observed that the surface strain distribution of the saturated specimens presented a similar trend with dry specimens at the same loading rates. Compared to the surface strains of the dry specimens with the same loading rate, it was found that the strain of the saturated specimens was lower than those of the dry specimens, suggesting that the viscosity provided by the internal free water of the saturated specimens hinders the development of strains in concrete.

## 4. Analysis of Material Internal Stress during Dynamic Loading

The dynamic strength test equipment of concrete can be divided into the split Hopkinson pressure bar (SHPB) and the rigidity test machine. The SHPB can provide a large loading rate, but the output is limited. The rigidity test machine can provide a significant output, but the loading rate is limited. It can be seen that no matter what kind of equipment is used to perform the dynamic test, the stress distribution of the specimen can be represented as shown in Figure 4. In Figure 4, *F* is the nominal external load applied to the concrete and can be recorded by the load sensor during the test, *F_i_* and *F_v_* are the inertial and viscous force acting on the specimen, and their directions are opposite to the acceleration and velocity. Therefore, under the influence of inertial and viscous force, the actual inner force *F_a_* exerting on the final failure surface can be expressed as
(1)$$F_{a}=F-F_{i}-F_{v}$$

Under static loading conditions, due to the long loading time, *F_i_* and *F_v_* of the material is minimal, and then Equation (1) can be simplified as
(2)$$F_{a}=F$$

With a high loading rate, the failure time is very short such that *F_i_* and *F_v_* cannot be ignored. Then the actual strength is less than the nominal strength of the material. Therefore, no matter what kind of test equipment is used for dynamic testing, if the measured nominal strength is directly used as the actual strength during the failure process, the dynamic capacity of the structure will be overestimated. Therefore, to accurately analyze the actual properties of concrete strength under different strain rates, the relationship between the nominal strength and the actual strength should be clarified. 

## 5. Single Degree of Freedom Model of Structural Dynamic Response

A single degree of freedom (SDOF) system is composed of a mass point and a spring is employed to analyze the influence of inertia during the dynamic test. For a concrete structure, as shown in Figure 5 and Figure 6, point *M* represents the mass of the concrete, and the spring stiffness *K* represents the elastic modulus of the concrete. It can be seen that *P* is the external load provided by the loading equipment on the loaded surface of concrete during the dynamic test. As the loading rate increases up to a specific value, under the influence of inertia, the load imposed on the spring will be different from the measured external load *P*.

As shown in Figure 6, *P*(*t*) is the dynamic load acting on the system, which is a linearly increasing function of the loading time, can be expressed as
(3)$$P\left(t\right)=\alpha t$$
where *α* is the loading rate and *t* is the loading time. To analyze the influence of the inertia on the system, the impact of damping on the system is ignored. Therefore, the governing equation of the system is expressed as [30]
(4)$$M\ddot{x}+Kx=P(t)\;\;x\leq x_{f}$$
where *x* is the displacement of the mass point, *x_f_* is the maximum value of the spring elongation. From Equation (4), the relationship between spring elongation and load time can be obtained as
(5)$$x=\frac{\alpha }{K}\left(t-\frac{1}{\omega }\sin\omega t\right)$$
where *w* is the natural frequency of concrete and can be expressed as
(6)$$\omega =\sqrt{K/M}$$

It can be seen from Equation (5) that when *t* = *t*_0_, *x* = *x_f_*, the external loading applied on the system is *αt*_0_, and the actual load on the spring can be shown as
(7)$$F_{sf}=Kx_{f}=P\left(t_{0}\right)-\frac{\alpha }{\omega }\sin\omega t_{0}$$
where *P*(*t*_0_) is the applied load of the test equipment when the concrete is failed and *F_sf_* is the actual load on the spring. When the loading rate is high, *P*(*t*_0_) is larger than *F_sf_*, and exhibits a rate effect. Then the DIF of the system can be expressed by *P*(*t*_0_)/*F_sf_* as shown in Equation (8)
(8)$$\mathrm{DIF}=\frac{\alpha t_{0}}{F_{sf}}=\frac{1}{1-\sin\left(\omega t_{0}\right)/\left(\omega t_{0}\right)}>1$$

It can be seen from Equation (8) that the influence of inertial force is inversely proportional to the loading time. The shorter the loading time is, the higher the influence of inertia on concrete strength. Figure 7 shows the relationship between DIF and loading rate, where *w* = 50 Hz [31]. It can be seen from Figure 7 that when the loading rate is less than 1/s, the spring will constantly reach equilibrium with the increase in the applied loading, and the value of *F_sf_* is close to the applied load *P*(*t*), resulting in a marginal effect of inertia on the dynamic strength of concrete. When the loading rate is larger than 1/s, the inertia force exerts a significant influence on the strength of concrete, and then, the concrete fails before it reaches the equilibrium state. Therefore, the DIF is significantly increased with the increase in the loading rate. However, as the strain rate increases to 10^1^/s, DIF attains 21 which is much larger than the dynamic nominal compressive strength of concrete. Thus, another model is needed to simulate the effect of inertia on the dynamic response of concrete.

## 6. Numerical Simulation of the Influence of Structure Inertia under Dynamic Loading

The above single-degree freedom and plane systems can only be used to qualitatively analyze the influence of inertia on the strength of elastic concrete materials under different loading rates. In order to further investigate the inertia influence of concrete during the dynamic tests, a 3D finite element model was established to simulate the dynamic compression response of concrete by ABAQUS software. As depicted in Figure 8, a cubic concrete with a side length of 150 mm was adopted in the test. A thick steel plate with a thickness of 15 mm was placed on top surface to apply dynamic, and the bottom surface was fixed on the test equipment. The interface between the upper steel plate and the specimen was simulated by the contact element.

The concrete damaged plasticity (CDP) model proposed by Lee [32] was applied to simulate the dynamic response of concrete. The compressive stress and tensile strength can be given by
(9)$$\sigma_{t}=(1-d_{t})E_{0}{\tilde{\varepsilon }}_{t}^{el}=(1-d_{t})E_{0}(\varepsilon_{t}-{\tilde{\varepsilon }}_{t}^{pl})$$
(10)$$\sigma_{c}=(1-d_{c})E_{0}{\tilde{\varepsilon }}_{c}^{el}=(1-d_{c})E_{0}(\varepsilon_{c}-{\tilde{\varepsilon }}_{c}^{pl})$$

The constitutive equation can be expressed as:(11)$$\sigma =(1-d)D_{0}^{el}:(\varepsilon -\varepsilon^{pl})=D^{el}:(\varepsilon -\varepsilon^{el})$$
where, *E*_0_ is the initial elastic modulus; *ε* is the total strain; ${\tilde{\varepsilon }}_{c}^{el}$, ${\tilde{\varepsilon }}_{t}^{el}$ represent the elastic strain of the concrete in compression and tension, respectively; ${\tilde{\varepsilon }}_{c}^{pl}$, ${\tilde{\varepsilon }}_{t}^{pl}$ represent the damage plastic strain of the concrete material in compression and tension, respectively; ${\tilde{\varepsilon }}_{0c}^{el}$, ${\tilde{\varepsilon }}_{0t}^{el}$ represent the non-damage elastic strain of the concrete in compression and tension, respectively; ${\tilde{\varepsilon }}_{c}^{in}$, ${\tilde{\varepsilon }}_{t}^{ck}$ represent the inelastic strain and cracking strain of the concrete material in tension and compression, respectively; $D_{0}^{el}$ is the non-destructive elastic stiffness of concrete, $D_{}^{el}$ is the material damage elastic stiffness, and *d* is the damage factor. The value of the CPD model parameters used in the concrete is listed in Table 2.

Different loading rates, 10^−5^/s~10^2^/s were considered in the test, where the loading rate of 10^−5^/s was chosen as a quasi-static loading condition. As shown in Figure 8, the loaded-end point A and the far loaded points B and C were selected to analyze the influence of inertia on the concrete under different loading rates. 

The stress cloud diagram at different loading stages is shown in Figure 9, where the elastic stage is shown in Figure 9a, the damage stage is shown in Figure 9b, and the failure stage is shown in Figure 9c. It can be seen from Figure 9 that the axial stress in the concrete was symmetrically distributed. In the elastic stage, the stress at each point in concrete was relatively small. Due to the restraint of the top and the bottom surface of concrete, a small range of stress concentration occurred at the corners, whereas the overall distribution of the concrete internal stress was relatively uniform. In the elastoplastic stage, the internal stress distribution of the concrete varied insignificantly with the increase in the loading rate, and the stress around the loaded-end was larger than the point far away from the loaded-end. With the increasing loading time, the stress concentration at the corners of the concrete was gradually decreased. In the failure stage, the stress distribution at each point of the concrete had little relationship with the loading rate. It can be seen that the stresses at points A, B, and C at different stages showed a decreasing trend, and the stress concentration at the concrete corners basically disappeared. It can be seen from the three failure stages that the stress near the loading end was always larger than the other parts of the specimen. 

Considering the stress recorded by load sensor is equal to the reaction force of loading plate, thence the DIF values of different strain rates can be expressed as
(12)$$DIF=\frac{f_{d}}{f_{s}}=\frac{\max(P_{RFd})/A}{\max(P_{RFs})/A}=\frac{\max(P_{RFd})}{\max(P_{RFs})}$$
where max(*P_RFd_*) is the maximum dynamic axial reaction force of the loading plate, max(*P_RFs_*) is the maximum static axial reaction force of the loading plate. The simulation results of concrete are listed in Figure 7. From the above calculation results of the DIF of the single degree freedom system, it can be seen that when the loading rate was less than 1/s, the variation of the DIF was small and the influence of concrete inertia was small. Numerical results showed that the increasing percentage of concrete strength is very small in a loading rate within the range of 10^−2^/s. As the loading rate increases up to 10^1^/s, the strength of concrete increases by about 24%. By comparing the numerical results of this paper with the CEB bilinear phenomenological empirical formula calculation results, it can be seen from Figure 7 that the DIF calculated by the CEB phenomenological empirical formula showed a significant increase with the increase in loading rate. In the calculated results of this paper, the DIF value under a low loading rate increased with the increase in the loading rate, but the increased level was minimal. When the loading rate was larger than 10^1^/s, the calculated DIF value had a significant improvement, which was close to the calculation result of the CEB phenomenological empirical formula. Therefore, it can be considered that when the loading rate is low (within the seismic rate range), the inertia effect on concrete can be ignored. Owing to the static strength of concrete having been adopted to study the dynamic response of concrete, it can be found in Figure 7 that the value of Actual DIF (the ratio of actual dynamic concrete strength to static strength during the numerical analysis) remained unaltered under different strain rates. Therefore, the dynamic strength obtained in the test could not reflect the true mechanical properties of concrete, and the rate sensitivity of concrete can be attributed to the influence of inertia and many other factors. It should be pointed out that the static mechanical parameters and the stress–strain constitutive relationship of concrete can be well applied to simulate the effect of inertia on the strength of concrete under dynamic loading.

## 7. Conclusions

In this paper, a single degree of freedom model was established to investigate the influence of inertia on DIF, and a rate-independent elastoplastic damage model of concrete was employed to simulate the actual strength and nominal strength of concrete under different strain rates. The conclusions can be drawn as follows:
(1)Under the influence of inertia, the stress near the loading end is larger than the far end, and the axial reaction force measured by the actuator is larger than the actual stress in concrete;(2)The effect of inertia on the nominal strength of concrete is related to the strain rate. When the strain rate is less than 1/s, the increase in DIF is small such that the inertial effect on the nominal concrete strength can be ignored, and the rate sensitive is mainly related to the viscous resistance of water in concrete, especially for wet concrete. When the strain rate is larger than 1/s, the nominal strength of concrete is greatly influenced by inertia;(3)In the structural dynamic equation, the effect of inertia on the dynamic response of concrete structures is considered. If the dynamic nominal strength of concrete is directly applied to evaluate the dynamic response of concrete structures, that may overestimate the influence of inertia on the capacity of concrete structures. However, whether the actual strength proposed in this study is appropriately applied, the structural analysis needs to be further studied.

## Figures and Tables

**Figure 1 materials-15-07278-f001:**
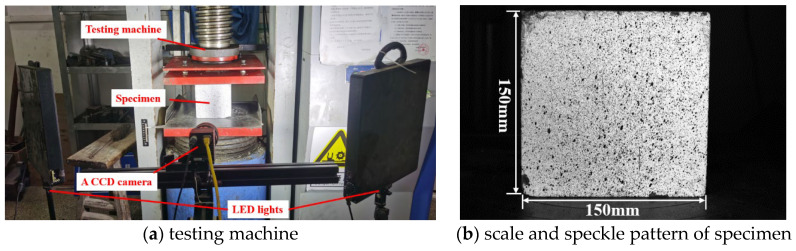
Test process.

**Figure 2 materials-15-07278-f002:**
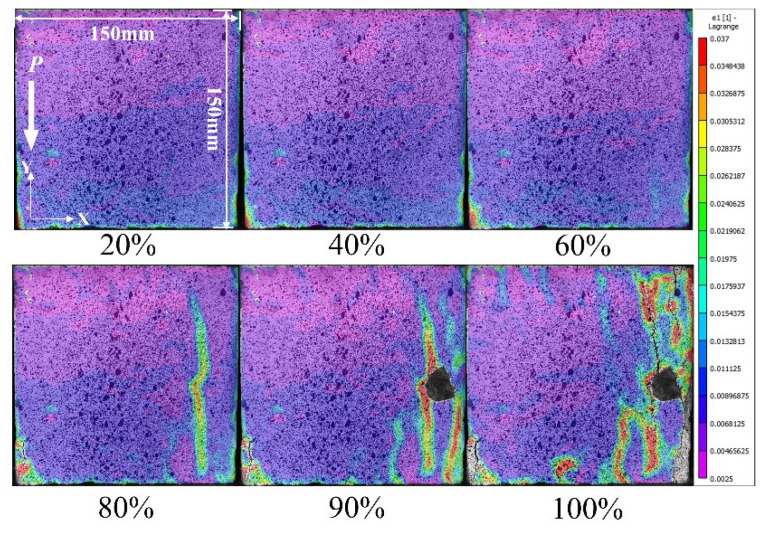
Principal strain distribution of D1 under different loading stages.

**Figure 3 materials-15-07278-f003:**
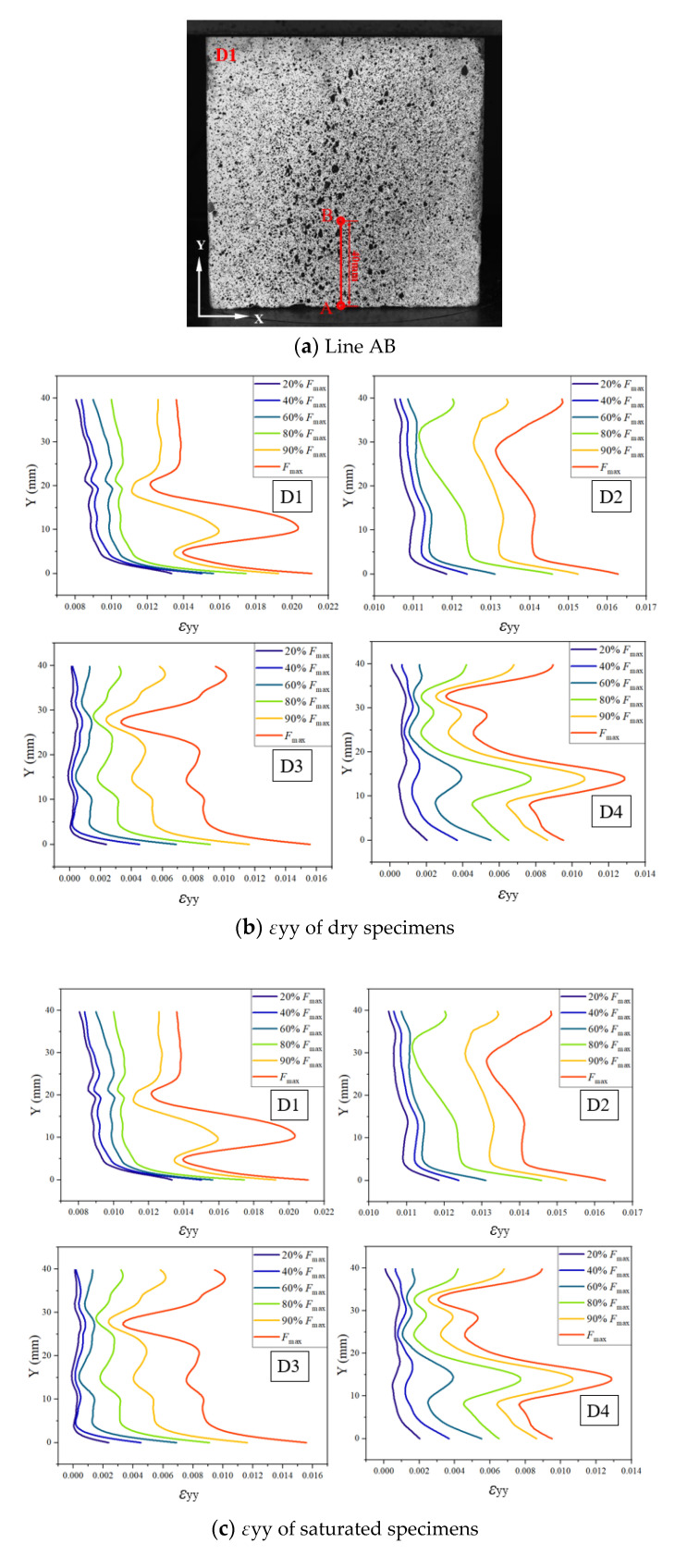
Variation of vertical strain of concrete with different moisture content. Therefore, under the effect of inertia, the stress of the loading end of the specimen was larger than the fixed end. To reveal the effect of inertia on the dynamic response of concrete, theoretical analysis and numerical simulation are needed.

**Figure 4 materials-15-07278-f004:**
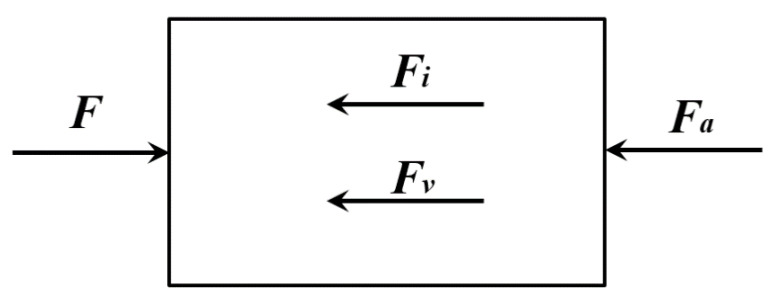
Schematic diagram of the force of the specimen.

**Figure 5 materials-15-07278-f005:**
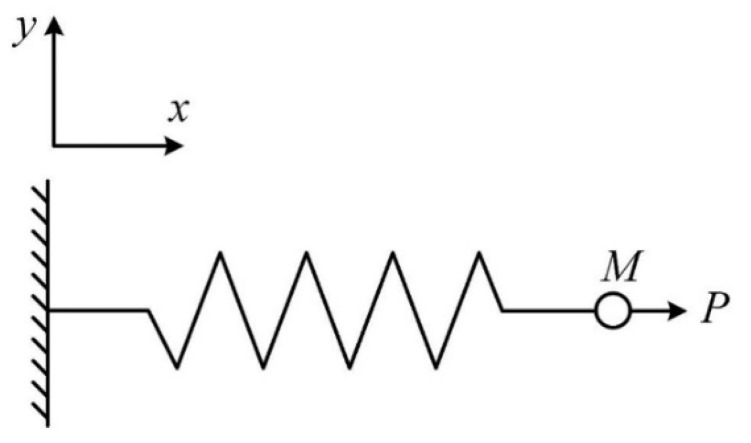
Single-degree-of-freedom system.

**Figure 6 materials-15-07278-f006:**
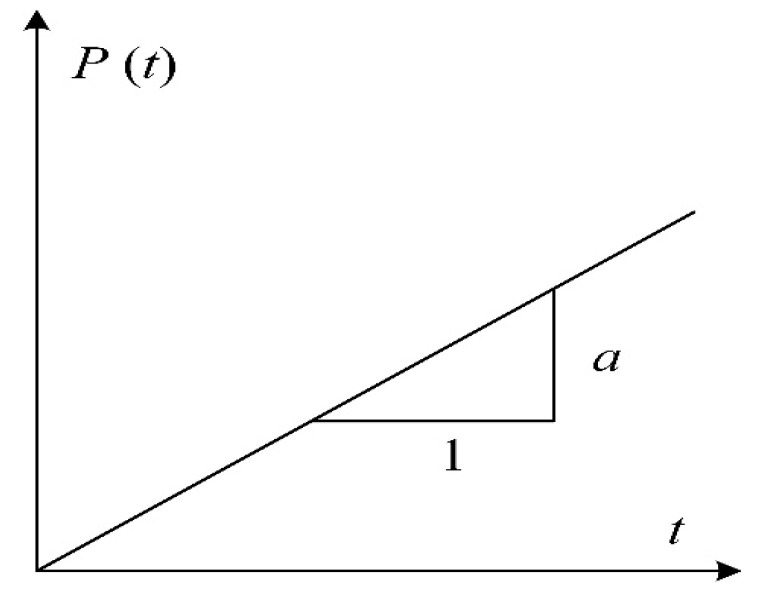
Relationship between dynamic load and loading time.

**Figure 7 materials-15-07278-f007:**
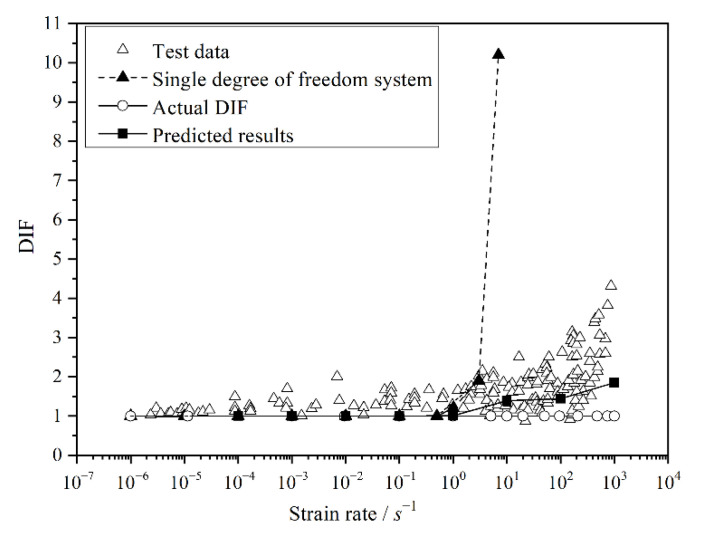
Simulation results of concrete dynamic compressive strength.

**Figure 8 materials-15-07278-f008:**
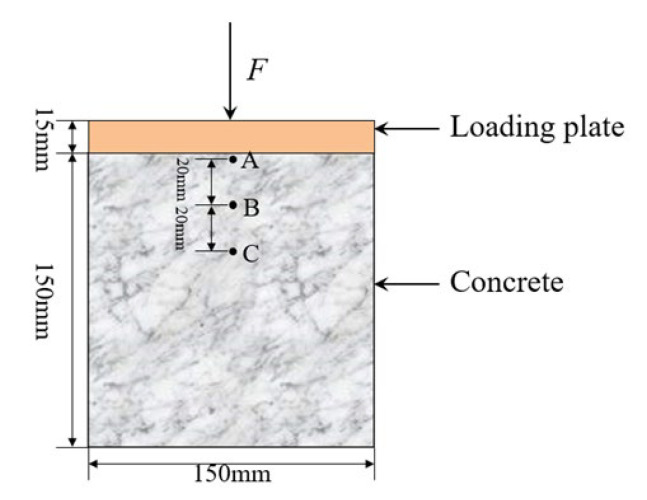
Simplified model of concrete dynamic tests.

**Figure 9 materials-15-07278-f009:**
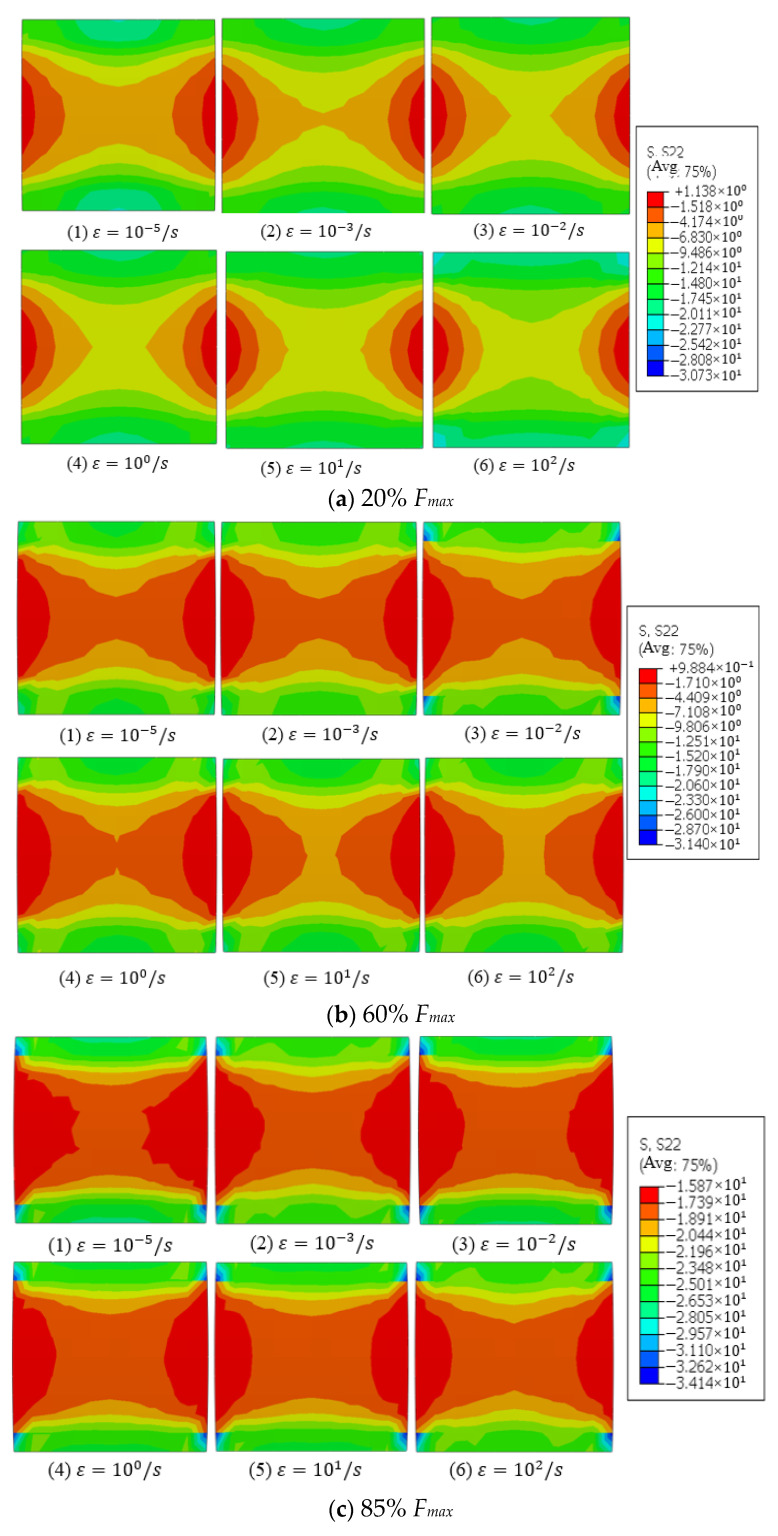
The stress cloud diagram under different loading stages.

**Table 1 materials-15-07278-t001:** Specimen details.

Number of Specimens	Moisture Content	Loading Rate	Frame per Second
D1	Dry	10^−5^/s	1
D2	Dry	10^−4^/s	2
D3	Dry	10^−3^/s	5
D4	Dry	10^−2^/s	10
S1	Saturated	10^−5^/s	1
S2	Saturated	10^−4^/s	2
S3	Saturated	10^−3^/s	5
S4	Saturated	10^−2^/s	10

**Table 2 materials-15-07278-t002:** Model parameters.

Density (kg/m^3^)	Elastic Modulus (GPa)	Poisson’s Ratio	Dilatancy Angle	Eccentricity	$f_{b0}/f_{c0}$	κ
2500	32.5	0.2	35	0.1	1.16	0.6667

## Data Availability

Not applicable.

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
