# Peer review of "Influence of Inertia on the Dynamic Compressive Strength of Concrete"

_materials, 2022, doi:10.3390/ma15207278_

Round 1
Reviewer 1 Report
The title of the manuscript is:"Influence of inertia on the dynamic strength of concrete".
The presentation of the study is relatively good. However, it is not clear from the title and the content of the paper which inertia is the subject of the investigation. Please provide a simple explanation for this issue.
Moreover, it is not clear why a one-degree freedom model is needed in the study, when a finite element model is also used. Please delete this chapter or convince the reader that it is important.
Author Response
Comment 1: The presentation of the study is relatively good. However, it is not clear from the title and the content of the paper which inertia is the subject of the investigation. Please provide a simple explanation for this issue.
Response 1: Many thanks. This paper aims to study the influence of inertia on the dynamic compressive strength of concrete material. Accordingly, revisions have been made as follows.
(1) On page 1 of the revised manuscript, the original title “Influence of inertia on the dynamic strength of concrete” has been replaced with “Influence of inertia on the dynamic compressive strength of concrete”.
(2) On page 2 of the revised manuscript, the original test “The rate sensitivity of concrete is closely related to the inertia and viscous effects.” has been replaced with “The rate sensitivity of concrete material is closely related to the inertia and viscous effects.”.
(3) On page 2 of the revised manuscript, the original test “concrete under different strain rate and verified with experimental results.” has been replaced with “concrete under different rates of compressive loading and verified with experimental results. The results obtained in this study indicated that the dynamic nominal strength of concrete obtained from the tests could not be directly used for structural analysis which may overestimate the effect of inertia on the dynamic response of the structure.”.
(4) On page 22 of the revised manuscript, the original text “The rate effect of concrete caused by the inertia can be simulated by the static stress-strain relationship. Therefore, the dynamic nominal strength of concrete obtained from tests could not be directly used for dynamic structural analysis.” has been changed to “In the structural dynamic equation, the effect of inertia on the dynamic response of concrete structures is considered. If the dynamic nominal strength of concrete is directly applied to evaluate the dynamic response of concrete structures, that may overestimate the influence of inertia on the capacity of concrete structures. However, whether the actual strength proposed in this study is appropriately applied to structural analysis needs to be further studied.”.
Comment 2: Moreover, it is not clear why a one-degree freedom model is needed in the study, when a finite element model is also used. Please delete this chapter or convince the reader that it is important.
Response 2: Many thanks. The one-degree freedom model we establish in this paper is from the perspective of structure, while the finite element model is established from the perspective of material. The two models are not contradictory but complement each other. Accordingly, revisions have been made as follows.
(1) On page 11 of the revised manuscript, the test “For a concrete structure,” has been added.
(2) On page 13 of the revised manuscript, the test “However, as the strain rate increases to 101/s, DIF attains 21 which is much larger than the dynamic nominal compressive strength of concrete. Thus, another model is needed to simulate the effect of inertia on the dynamic response of concrete.” has been added.
(3) On page 13 of the revised manuscript, Fig. 7 has been redrawn.
(4) On page 22 of the revised manuscript, the original text “The rate effect of concrete caused by the inertia can be simulated by the static stress-strain relationship. Therefore, the dynamic nominal strength of concrete obtained from tests could not be directly used for dynamic structural analysis.” has been changed to “In the structural dynamic equation, the effect of inertia on the dynamic response of concrete structures is considered. If the dynamic nominal strength of concrete is directly applied to evaluate the dynamic response of concrete structures, that may overestimate the influence of inertia on the capacity of concrete structures. However, whether the actual strength proposed in this study is appropriately applied to structural analysis needs to be further studied.”.
Reviewer 2 Report
Dear Authors,
Thank you for your manuscript.
In addition to the comments in the pdf (please see attached), I have few general concerns:
1. Please specify scale in figure 1.
2. I suppose Digital Image Correlation technology has been used. Not digital image.
3. Please specify the water content in dry and saturate conditions. I am not sure what the difference is between dry 1-4 or saturated 1-4.
4. How thick was the steel plate that was placed on the top of the concrete block?
5. For Figure 10, please ensure the scale is same for comparison purposes among elastic, failure and damage mode.
6. What is elastoplastic stage in Figure 10?
Thank you.

Author Response
Comment 1: Please specify scale in figure 1.
Response 1: Many thanks. The scale of the specimen and the labels of the testing equipment have been added in figure 1.
Comment 2: I suppose Digital Image Correlation technology has been used. Not digital image.
Response 2: Many thanks. The clerical error has been corrected.
Comment 3: Please specify the water content in dry and saturate conditions. I am not sure what the difference is between dry 1-4 or saturated 1-4.
Response 3: Many thanks. The specimens adopted in this paper can be divided into dry and saturated specimens. For dry specimens, the cured specimens were placed in an oven at the temperature of 90℃ for around 15 days until the weight of the specimens is not changed with the drying time. However, for saturated specimens, the specimens were placed in water until the weight of the specimens are not changed with the immersion time. Accordingly, on page 5 of the revised manuscript, the original text “For dry specimens, the specimens were placed in a dryer for 15 days until the weight no longer changed and the evaporation of internal free water was considered completed. For saturated specimens, the specimens will be automatically filled in the vacuum filling machine. After the specimens were dry and saturated, sprayed irregular speckles on their surface.” has been replaced with “The specimens adopted in this paper can be divided into dry and saturated specimens. For dry specimens, the cured specimens were placed in an oven at the temperature of 90℃ for around 15 days until the weight of the specimens is not changed with the drying time. However, for saturated specimens, the specimens were placed in water until the weight of the specimens are not changed with the immersion time. After the specimens were dry and saturated enough, irregular speckles were sprayed on the testing surface.”.
Comment 4: How thick was the steel plate that was placed on the top of the concrete block?
Response 4: Many thanks. The thick of the steel plate that was placed on the top of the concrete block is 15 mm and the detailed size of the specimen is listed in Figure 8.
Comment 5: For Figure 10, please ensure the scale is same for comparison purposes among elastic, failure and damage mode.
Response 5: Many thanks. In Figure 10, the scale of the stress for specimens under different loading stages are the same.
Comment 6: What is elastoplastic stage in Figure 10?
Response 6: Many thanks. The elastic, damage, and failure stage present in Figure 10 are corresponding to 20%, 60%, and 85% of the compressive strength, respectively. Accordingly, the elastic, damage, and failure stage present in Figure 10 are replaced with 0.20 Fmax, 0.60 Fmax, and 0.85 Fmax, respectively.
More response to your comments,please see Revised Manuscript.
Reviewer 3 Report
In this study, 5 digital image technology was applied to study the strain variation of dry and wet concrete under 6 different loading rates. I believe that the article will be published if the authors pay attention to the following points:
1) The current contribution of the research to the literature and its impact should be emphasized. It should not be just about research results.
2) According to which standard did you make the test setup? Please specify.
3) What is the number of repetitions of the specimens?
4) If there is one of each specimen, how did you ensure the accuracy of the imaging?
5) The conclusion section should be expanded.
Author Response
Comment 1: The current contribution of the research to the literature and its impact should be emphasized. It should not be just about research results.
Response 1: Many thanks. Accordingly, revisions have been made as follows.
(1) On page 2 of the revised manuscript, the text “The results obtained in this study found that the dynamic nominal strength of concrete obtained from the tests could not be directly used for structural analysis which may overestimate the effect of inertia on the dynamic response of the structure.” has been added.
(2) On page 22 of the revised manuscript, the original text “The rate effect of concrete caused by the inertia can be simulated by the static stress-strain relationship. Therefore, the dynamic nominal strength of concrete obtained from tests could not be directly used for dynamic structural analysis.” has been changed to “In the structural dynamic equation, the effect of inertia on the dynamic response of concrete structures is considered. If the dynamic nominal strength of concrete is directly applied to evaluate the dynamic response of concrete structures, that may overestimate the influence of inertia on the capacity of concrete structures. However, whether the actual strength proposed in this study is appropriately applied to structural analysis needs to be further studied.”.
Comment 2: According to which standard did you make the test setup? Please specify.
Response 2: Many thanks. The test setup according to the standard “Test code for hydraulic concrete (SL/T 352-2020)”. Accordingly, on page 5 of the revised manuscript, the test “According to the test code for hydraulic concrete (SL/T 352-2020)” has been added.
Comment 3: What is the number of repetitions of the specimens?
Response 3: Many thanks. 8 specimens with dimensions of 150 × 150 × 150 mm were adopted in the tests to study the strain variation of dry and saturated concrete under different loading rates. Therefore, one specimen of each test condition was sprayed with irregular speckles before testing. Accordingly, on page 5 of the revised manuscript, the original text “In order to study the influence of inertia on the dynamic strength of concrete, two types of concrete specimens were cast, dry and saturated, both were 150 * 150 * 150 mm in size of same proportion.” has been replaced with “According to the test code for hydraulic concrete (SL/T 352-2020), 8 specimens with dimensions of 150 × 150 × 150 mm were adopted in the tests to study the strain variation of dry and saturated concrete under different loading rates. Therefore, one specimen of each test condition was sprayed with irregular speckles before testing.”.
Comment 4: If there is one of each specimen, how did you ensure the accuracy of the imaging?
Response 4: Many thanks. The VIC-2D system was adopted in the test with the accuracy of 10 uε and developed by the Correlated Solutions INC, USA. Before testing, the accuracy of this VIC-2D system was verified by the strain gauge with an accuracy of 0.5 uε. Tests showed that the error between DIC technology and the strain gauge was within 3%. Accordingly, on page 6 of the revised manuscript, the test “The VIC-2D system was adopted in the test with the accuracy of 10 uε and developed by the Correlated Solutions INC, USA. Before testing, the accuracy of this VIC-2D system was verified by the strain gauge with an accuracy of 0.5 uε. Tests showed that the error between DIC technology and the strain gauge was within 3%.” has been added.
Comment 5: The conclusion section should be expanded.
Response 5: Many thanks. Accordingly, on page 22 of the revised manuscript, the original text “The rate effect of concrete caused by the inertia can be simulated by the static stress-strain relationship. Therefore, the dynamic nominal strength of concrete obtained from tests could not be directly used for dynamic structural analysis.” has been changed to “In the structural dynamic equation, the effect of inertia on the dynamic response of concrete structures is considered. If the dynamic nominal strength of concrete is directly applied to evaluate the dynamic response of concrete structures, that may overestimate the influence of inertia on the capacity of concrete structures. However, whether the actual strength proposed in this study is appropriately applied to structural analysis needs to be further studied.”.